# Occurrence and Toxicological Risk Evaluation of Organochlorine Pesticides from Suburban Soils of Kenya

**DOI:** 10.3390/ijerph16162937

**Published:** 2019-08-15

**Authors:** Teresiah M. Mungai, Jun Wang

**Affiliations:** 1College of Marine Sciences, South China Agricultural University, Guangzhou 510642, China; 2Key Laboratory of Aquatic Botany and Watershed Ecology, Wuhan Botanical Garden, Chinese Academy of Sciences, Wuhan 430074, China; 3University of Chinese Academy of Sciences, Beijing 100049, China; 4Sino-Africa Joint Research Center, Chinese Academy of Sciences, Wuhan 430074, China

**Keywords:** cancer risk, concentrations, contamination, dichlorodiphenyltrichloroethane (DDT), hexachlorocyclohexanes (HCH), organochlorine pesticides (OCP)

## Abstract

The use of organic chemicals in agriculture and manufacturing has raised concerns about the dangers of organochlorine pesticides (OCPs) in the environment. By examining OCPs occurrence in the suburban soils from Kenya, this study revealed the distribution, concentrations, and the threat posed to the environment and human health. A gas chromatography electron capture detector was used to test the pesticides. The hexachlorocyclohexane (HCH) and dichlorodiphenyltrichloroethane (DDT) studied in soils of Kapsabet, Voi, and Nyeri towns showed concentrations ranging from 0.03–52.7, 0.06–22.3, and 0.24–24.3 ng/g respectively. The highest concentration of HCHs was in Kapsabet (0.03–48.1 ng/g), whereas the highest DDTs concentration was in Voi (n.d.–15.5 ng/g). Source identification revealed OCPs pollution originated from recent usage of DDT pesticides to control insect-borne diseases and from the use of lindane in agriculture. Correlation test revealed that total organic carbon influenced the presence of pesticides in the soils. The enantiomeric ratios of α-HCH/γ-HCH were <3 indicating the use of lindane while the ratios of DDE/DDT were <1 suggesting recent input of DDT. The cancer risk assessment showed values close to the set risk level of 10^−6^, suggesting the likelihood of exposure to cancer was not low enough, and control measures need to be established.

## 1. Introduction

Environmental pollution from organic compounds, including organochlorine pesticides (OCPs), is continually increasing with rapid urbanization and industrialization activities [1]. Health risks from OCPs, including dichlorodiphenyltrichloroethane (DDT) and hexachlorocyclohexane (HCH), are a growing concern. These compounds are of significance as they can bioaccumulate and stay for more extended periods in the environment [2,3]. Upon exposure to OCPs, health complications including organ failure, disruption of the endocrine system, and even cancerous tumors may occur [4]. In the Eastern part of Africa, the usage of OCPs has been extensive over the years, leading to their accumulation in the soil [5]. According to the Stockholm Convention, DDT and HCH, are amongst the 12 most persistent organic pollutants and are of global concern [6]. HCH chemicals exist in different forms called isomers including α, β, γ, and δ-isomers. The HCH isomers are the most prevalent halogenated organic insecticides [7]. In the past, HCHs were used for crop protection [8] and the treatment of parasites in animals [9]. However, due to their persistent nature, toxicity, and tendency to bioaccumulate, HCHs were forbidden from use [10]. Nonetheless, despite this restriction, HCHs are still present in the environment [11], and they stand to be hazardous to human health and the environment [12,13]. 

DDT, on the other hand, is an organochlorine compound that does not readily degrade in the environment. DDT is released into the environment through the application of technical grade DDT, which consists of different isomeric composition including the following: p,p′-DDT 77.1%; o,p′-DDT 14.9%; p,p′-DDD 0.3%; o,p′-DDD 0.1%; p,p′-DDE 4.0%; o,p′-DDE 0.1%. Previously, DDT was extensively used to prevent malaria and to control agricultural pesticide in crop protection [14]. DDT then accumulates in the soil and sediments and can be transported from its point of application via atmospheric transport or soil erosion [15]. Owing to DDTs relatively high persistence in the environment, residues from its past use may be detectable [16]. A previous report stated that DDT bioaccumulates in human and animal tissue and compromises the immune system disrupting the reproductive and nervous systems [17]. Furthermore, the United States Environmental Protection Agency (US EPA) has named DDT as a possible cancer-causing agent leading to the adoption of legislative measures to control and manage POPs in the environment [18]. 

In Africa, pesticides have previously been used for controlling agricultural pests and insect-vectors in agriculture and public health, respectively [19]. In Kenya, for instance, the ban on DDTs pesticides importation was reported in 1985 [20]. However, despite the restrictions imposed, much of the contamination levels in the environment remains unknown [21]. A previous study conducted in Kenya showed that elevated levels of OCPs in soil (∑HCHs 5.5, ∑DDTs 25.99 ug/kg) posed both ecological and human health risk. The study also reported that OCP pollution was as a result of new inputs of OCPs, including DDTs in the environment [22]. In Uganda, a study reported OCP contamination in soil with a mean of 59.00 ug/kg, and the concentration levels were linked to the use of prohibited insecticides and the tainting of pesticides [5]. Similarly, in Shanghai, ∑HCHs n.d.–0.38 ng/g and ∑DDTs 0.77–247.45 ng/g were reported to be the most dominant compounds, and their presence was attributed to both recent and historical applications [23].

In Kenya, soil contamination has, over the past years, become widespread, with the increased use of chemicals, especially in agriculture, and manufacturing sectors. These compounds are important for modern-day life, but, they pose potential risks to human health if not closely monitored. This study was thus formulated to examine OCP contamination in three major suburban towns of Kenya. The study aimed and to gain more insight into the intensities and distribution of HCHs and DDTs in soils; to evaluate the possible sources of the DDTs and HCHs compounds and to determine the pollution status of DDTs and HCHs concentrations in soils, and, more importantly, the risk posed to human health using the incremental cancer risk (ILCR) method.

## 2. Materials and Methods 

### 2.1. Sample Sites

The sampling sites from this study focused on locations influenced by anthropogenic activities from nearby urban and agricultural areas (Figure 1). The towns are also characterized by a high population density coupled with improvements in infrastructure and the rise of industries. The first study site was Nyeri, a town situated in the central region of Kenya. The area is well known for farming of coffee and tea and is also home to several manufacturing industries, including a soft drinks bottling plant, leather products processing plant, tea, and coffee manufacturing industries. Voi, on the other hand, is a town located on the Nairobi–Mombasa highway close to the Tsavo East National Park. In this area, agronomy is common comprising of large sisal farms. The third study site is Kapsabet, the capital of Nandi County. Maize and tea are conventional crops grown. The town also has tea factories, including the KTDA Chebut Tea Factory and Kipchabo Tea Factory.

Anthropogenic pollution sources in the area include intense farming activities where pesticides and commercial fertilizers are commonly used. The inappropriate disposal of municipal waste matter is also common due to the lack of proper waste management facilities. There is also pollution from vehicular emissions, especially from the heavy commercial vehicles on the highway transporting industrial products.

### 2.2. Sampling and Pretreatment

Soil samples were collected in three different counties in Kenya as a representation of different parts of the country, providing the possibility for comparison between sites. Sampling concentrated on areas influenced by anthropogenic activities in the adjoining urban centers and agricultural areas. A total of 52 soil samples were collected; 20 samples were obtained from Kapsabet town, nine samples from Voi town, and 23 samples were obtained from Nyeri town. At each sampling point, the upper depth of the soil (0–20 cm) was sampled. In the laboratory, all the soil collected was put in paper bags and preserved at −20 °C to avert any adverse changes. The soil samples were then dried using a freeze-drier for about 48 h, ground to fine powder, and sifted through a 60-mesh nylon sieve to remove the coarse debris. 

### 2.3. Sample Extraction and Analysis

The standard solution including β-HCH > α-HCH > δ-HCH > γ-HCH, p,p′-DDT, p,p′-DDE, p,p′-DDD, and o,p′-DDT were acquired from AccuStandard (New Haven, CT, USA). Extraction was done according to a similar procedure [22] where, the soil sample (1 g) was thoroughly mixed with C18 (3 g) (Silicycle, Inc., Quebec, QC, Canada) for about 5 min. The blended matter was then conveyed into a 10 mL solid-phase extraction cartridge column. The cartridge column was packed with Na_2_SO_4_ (0.5 g) (Sino Pharm Chemical Reagent Co., Ltd., Shanghai, China), florisil (1 g, 60–100 mesh) (Beijing Yizhong Chemical Plant, Beijing, China) acidic silica gel (1 g) (Qingdao Haiyang Chemical Co., Qingdao, China), and copper powder (0.5 g) (Sino Pharm Chemical Reagent Co., Ltd., Shanghai, China). These constituents were arranged from the bottom to the top, respectively. Using dichloromethane (15 mL), (Fisher Scientific, Waltham, MA, USA), the analytes were then eluted into glass bottles. The eluent collected was further dried under a smooth flow of nitrogen gas and the residues re-solubilized with normal hexane (100 uL) (Sino Pharm Chemical Reagent Co., Ltd., Shanghai, China). Lastly, the constituents were analyzed using a gas chromatography-electron capture detector (GC-ECD). For the measurement of the total organic carbon (TOC), the soil sample (50 mg) was tightly bound by a tin foil and then placed in the TOC analyzer (SSM-5000A, Shimadzu, Kyoto, Japan). 

### 2.4. Instrumental Analysis

A gas chromatography-electron capture detector (GC-ECD, Agilent 7890B) was used to examine the OCPs in the soil samples. A special tube column HP-5MS (Agilent Technology, 30 m by 0.25 mm i.d by 0.25 um) was used to separate the analytes of interest. The GC-ECD was run in splitless method, and 1 uL of the excerpt was introduced into the GC-machine to extricate the OCPs. The helium carrier gas was used at a frequency of 1.0 mL/min. The temperature of the injector was held at 280 °C, while the gauge temperature was retained at 300 °C. The kiln temperature was first set at 80 °C (1 min), then to 190 °C (2 min) at 15 °C/min, 220 °C (5 min) at 8 °C/min, and lastly to 300 °C (7 min) at 10 °C/min. 

### 2.5. Quality Control 

Before analysis, we processed a routine blank sample to ensure that no interference occurred. The detection limit of the method ranged from 0.001 to 0.025 ng/g, and the limit of quantification was 0.155–0.167 ng/g. The parameters of detection for the OCPs were equated to a signal-to-noise ratio of 3:1. The average retrievals of OCPs varied from 80–102% with relative standard deviation (RSDs) from 5% to 10%. The results of the correlation coefficients derived for the standard curves showed values greater than 0.995. The OCPs were calculated by assessing the area under each of the sample curves and then comparing it with the area under the standard peaks. 

### 2.6. Carcinogenic Risk Assessment 

The incremental cancer risk ILCR is a model that characterizes the incremental possibility of a person to develop cancer during his lifespan [24]. Exposure to a probable carcinogen compound was classified according to three major pathways; through ingestion, skin contact, and inhalation for children, adolescent, and adults [25]. Children are defined as individuals from 0–10 years, adolescents are defined as individuals from ages 11–18 years, while adults are defined as individuals from 19–70 years. We evaluated the ILCR of communities residing in urban districts of Nyeri, Voi, and Kapsabet affected by exposure to DDTs and HCHs. The following equations were used [25,26,27].
(1)$${\mathrm{ILCRs}}_{\mathrm{ingestion}}=\frac{C_{\mathrm{soil}}\times \left({\mathrm{CSF}}_{\mathrm{ingestion}}\times \sqrt[{3}]{\mathrm{BW}/70}\right)\;\times {\mathrm{IR}}_{\mathrm{soil}}\times \mathrm{EF}\times \mathrm{ED}}{\mathrm{BW}\times \mathrm{AT}\times \mathrm{CF}}$$
(2)$${\mathrm{ILCRs}}_{\mathrm{dermal}}=\frac{C_{\mathrm{soil}}\times \left({\mathrm{CSF}}_{\mathrm{dermal}}\times \sqrt[{3}]{\mathrm{BW}/70}\right)\times \mathrm{SA}\times \mathrm{FE}\times \mathrm{AF}\times \mathrm{ABS}\times \mathrm{EF}\times \mathrm{ED}}{\mathrm{BW}\times \mathrm{AT}\times \mathrm{CF}}$$
(3)$${\mathrm{ILCRs}}_{\mathrm{inhalation}}=\frac{C_{\mathrm{soil}}\times \left({\mathrm{CSF}}_{\mathrm{inhalation}}\;\times \sqrt[{3}]{\mathrm{BW}/70}\right)\times {\mathrm{IR}}_{\mathrm{air}}\times \mathrm{EF}\times \mathrm{ED}}{\mathrm{BW}\times \mathrm{AT}\times \mathrm{PET}}$$
where $C_{\mathrm{soil}}$ refers to the contaminant concentration in soil (mg/kg); CSF represents the carcinogenic slope factor (1/(milligram/kg/day)); BW is the typical body mass (13.95 kg for child, 46.75 kg for adolescent, and 58.75 kg for adults). IR soil is the ingestion frequency of soil (mg/day) where the rate was (200 for child and 100 for both adolescent and adulthood); EF is the exposure frequency (350 days/annum); ED is the exposure period (6 years, 14 years, and 30 years for child, adolescent, and adult, correspondingly); LT is the lifetime (72 years); AT is the average period given in days (LT × 365 days); SA is the surface area of skin that touches the soil (2800 cm^2^/day); FE represents the fraction of dermal exposure ratio to soil (0.61); CF is the conversion factor (1 × 10^−6^ mg/kg); AF is the adherence factor of soil to the skin (0.2 mg/cm for child and adolescent and 0.07/mg/cm for adulthood); ABS is the dermal adherence factor (0.13); (chemical-specific values). IR_air is the inhalation rate (10.9 m^3^/day for childhood, 17.7 m^3^/day for adolescent, and 17.5 m /day for adulthood); and PET is the particle emission factor (1.36 × 10^9^) m^3^ /kg) for inhalable particles PM10.

The carcinogenic risk was grouped into three different age groups due to some exposure parameter, including the difference in body mass, the rate of ingestion, and inhalation rates. The total risk posed by OCPs in the study sites was obtained by the sum of risks of ILCRs for the three exposure pathways was calculated. The parameters involved are provided in Table 1.

### 2.7. Statistical Analysis

The data analysis was examined using the Statistical Package for the Social Sciences (version 23.0, Chicago, IL, USA) and Microsoft Excel 2013. Pearson correlation test was employed to examine the link between the OCPs and between TOC and OCP concentrations at the *p* ≤ 0.05 degree of confidence. Probable pollution sources were determined using principal component analysis (PCA). PCA is a tool for predicting sources of pollution. It groups’ data into different interrelated modules based on their component loadings.

## 3. Results and Discussion

### 3.1. Concentrations of DDTs and HCHs in Soils of Kenya

The concentration profiles of DDTs and HCHs in soils from Kapsabet, Nyeri, and Voi were calculated (Table 2). The concentrations were in the range of 0.03–52.7, 0.24–24.3, and 0.06–22.4 ng/g for Kapsabet, Nyeri, and Voi, respectively. The highest OCP levels were in Kapsabet Town, with an average value of 6.97 ± 14 ng/g. Agriculture is the main economic activity in Kapsabet town with plantations of tea and maize in the surroundings. Additionally, the town has tea factories, including the KTDA Chebut Tea Factory and Kipchabo Tea Factory. The high levels of OCPs could have arisen from chemical and pesticide uses in the tea farms. The prevalent OCPs in the three study regions were from the HCH-families. The total HCHs concentrations (sum of α-HCH, β-HCH, γ-HCH, and δ-HCH) varied from, 0.03–48.1, 0.06–6.86, and 0.24–4.72 ng/g for Kapsabet, Voi, and Nyeri, respectively, while those for DDTs ranged from n.d.–19.6, n.d.–15.5 ng/g, and n.d.–4.68 ng/g in Nyeri, Voi, and Kapsabet, respectively. The higher ratios of HCHs could be from the widespread use of HCH pesticides for agricultural purposes [12]. Moreover, a previous study done in Kenya reported the sustained use of chemicals containing HCHs despite them being prohibited [28]. The concentration of HCHs isomers in Kapsabet followed the sequence β-HCH > α-HCH > δ-HCH > γ-HCH. While in Nyeri and Voi, HCHs isomers followed the series β-HCH > γ-HCH > δ-HCH > α-HCH. In aged environmental samples, α-HCH and γ-HCH can be changed to β-HCH [29], which is more persistent in the environment compared to other HCH isomers [30]. p,p′-DDT, p,p′-DDE, p,p′-DDD, and o,p′-DDT were all present in the Nyeri soil samples. Nyeri County is renowned for tea and coffee production; therefore, the presence of DDT metabolites in the soil could be associated with the use of pesticides containing DDT in the tea and coffee farms. In Kapsabet town, trace levels of p,p′-DDE and o,p′-DDT (∑DDTs 0.85 ± 1.3 ng/g) were detected. These concentration levels were attributed to the application of pesticides in agriculture to curb crop pesticides and from soil erosion in areas where they had been used. The concentrations of p,p′-DDE and p,p′-DDD in Voi were linked to the agricultural activity in the large sisal farms. It has been previously documented that the direct application of pesticides in agriculture is a major source of DDTs in the soil, which could explain the concentration levels in this study area [31]. The peak ∑DDTs concentration was in Voi town with an average of 3.18 ± 5.6 ng/g. In Voi town, the use of DDT pesticides in controlling pests and diseases in the sisal fields could be a contributing factor. Amongst the DDT metabolites, o,p′-DDT dominated with values of 1.52 ± 3.2 ng/g in Voi, 0.41 ± 0.95 ng/g in Nyeri, and 0.37 ± 0.90 ng/g in Kapsabet. o,p′-DDT is a metabolite of DDT this could imply that the use of technical DDT pesticides could have led to an upsurge of DDT metabolites in the soil of Kenya. Overall, the concentrations of OCPs obtained were compared with the Dutch standard quality soil values (Table 2). The levels of both the HCHs and DDTs from our study showed values within the set target for uncontaminated soil [32].

In this study, the level of ∑HCHs, 6.12 ± 13 ng/g, was close to those reported previously in Kenya (∑HCHs 5.5 ng/g) [22], while the ∑DDTS 3.18-5.61 ng/g reported lower concentration levels (∑DDTs 25.99 ng/g) [22]. Similarly, in Shanghai, (∑HCHs n.d.–10.38 ng/g, ∑DDTs 0.77–247.45 ng/g) [23], ∑HCHs were found to be lower than those reported in this study area while the ∑DDTs were reported to be higher than this study. When compared to those of Uganda (∑OCPs 59.0 ug/kg) [5], China (∑HCHs n.d–8.96 ng/g, ∑DDTs n.d.–94.07 ng/g) [33], Hanoi Vietnam (∑HCHs 0.05–20.57 ng/g, ∑DDTs 0.02–171.83 ng/g) [34], and Poland (∑HCHs 0.36–110 ng/g, ∑DDTs 4.3–2400 ng/g) [35], the OCP contamination levels in this study reported lower concentrations.

Previously, the proportions between OCPs compound and their metabolites have been used to identify pollution sources. Thus, to determine the sources of HCHs, the isomer ratio of α-HCH and γ-HCH was used. When the ratio α-HCH/γ-HCH is less than 1, this indicates that the HCHs sources could have emanated from the use of lindane. Alternatively, if the ratio is between 3 and 7, this could be linked to the use of technical HCHs [36]. Technical HCH is primarily composed of α-HCH (60–70%), β-HCH (5–12%), γ-HCH (10–15%), δ-HCH (6–10%), and lindane γ-HCH (99%) [37]. In Kapsabet, Voi, and Nyeri, the ratios of α-HCH/γ-HCH were 2.51, 0.66, and 0.51, respectively. These values did not fall in the range of 3 and 7 (Table 2) implying the possible usage of lindane [38]. Lindane is a widely used chemical for seed treatment in Kenya, hence explaining its abundance in the soil [21]. 

The principal constituents of p,p′-DDT is p,p′-DDE and p,p′-DDD. Their ratios indicate current or historical use of DDT [39]. If the ratio of DDE/DDT is high, this could mean that the DDT in the soil is from past input. However, if the ratio DDE/DDT is low (<1), this indicates recent input [40]. The ratios of DDE/DDT from our study was generally less than in all the study sites, which inferred the new contribution of DDT. This result was consistent with research in Kenya, which reported the use of DDTs for health preventive measures, primarily due to its effectiveness in controlling malaria [21,22]. Additionally, the physicochemical properties of DDT, including its lower water solubility and high affinity for lipids also aid in their long-term residues in soil [41].

The composition percentage of total OCPs detected in soils from Kenya were illustrated in Figure 2. The concentration values of HCHs accounted for 69.4% while DDTs accounted for 30.6% of the total OCPs. In Nyeri, all four DDTs and four HCHs contaminants tested were identified (Figure 2a). This inferred use of different pesticides containing DDTs and HCHs in the soil. Nyeri is also well known for the farming of coffee and tea. Therefore, field pesticides and herbicides may be used for the control of insect infestation and diseases [42]. Overall, β-HCH showed relatively higher values in all the study sites, including Kapsabet (69%), Nyeri (25%), and Voi (32%), compared to the other OCPs (Figure 2). Previous studies reported that β-HCH, in comparison to different HCH isomers, has a stable molecular structure. This renders β-HCH, relatively stable, and highly persistent in the soil [30]. In Voi, o,p’-DDT (24%) showed significantly high levels compared to the other study sites (Figure 2c). The o,p’-DDT levels were linked to the need for the use of specific pesticides to regulate plant parasites.

### 3.2. Sources of DDTs and HCHs in Soils

Principal component analysis (PCA) was employed to define the potential origin of HCHs and DDTs based on their interrelated patterns. The percentage variance for the three counties analyzed was explained using four, three, and, two eigenvector factors for Nyeri, Kapsabet, and Voi, respectively (Table 3). In the case of, Nyeri, the first principal component factor (PC1), with a variance of 28.6%, was highly associated with β-HCH, γ-HCH, and o,p′-DDT. β-HCH was the dominant compound in this area, possibly due to the conversion of α-HCH and γ-HCH by microorganisms to β-HCH [30]. The second component (PC2), with a variance of 22.9% had high positive loading for p,p′-DDD, and p,p′-DDT. With reference to the results of the isomer ratios stated before, DDT primarily originated from fresh inputs inferring the recent use of DDTs. In Nyeri, the presence of DDTs may be as a result of farming activities from the coffee and tea plantations. The third component (PC3) had a variance of 15.9% and showed high loading scores of α-HCH and δ-HCH, indicating they were from a similar source. The fourth component matrix (PC4), with a variance of 14.7%, showed positive loading scores of p,p′-DDE, inferring the use of crop-specific pesticides containing DDT pesticide, which transforms into to DDE [43]. In Kapsabet, the first component explained the variance of 36.2% and comprised of α-HCH, β-HCH, and γ-HCH, indicating the pre-decomposition of the three HCHs isomers. The second component, with a variance of 26.7%, showed loading scores of p,p′-DDT, and o,p′-DDT while the third component with a variation of 17.7% showed loading scores of δ-HCH, p,p′-DDE, and p,p′-DDD. In Africa, malaria remains an epidemic, and as a result, DDT, despite being discontinued for other uses, is still being used for the control of malaria [44]. The application of DDT pesticides in malaria control can, therefore, be ascertained to be a significant source of DDTs in this study [21]. In Voi, two components were derived. The first component factor (PC1) explained the variance of 66.6% while the second (PC2) explained 12.7% of the total variance. The first component matrix had strong correlations α-HCH, β-HCH, γ-HCH, δ-HCH, p,p′-DDE, and o,p′-DDT. The second component showed loading scores of α-HCH, β-HCH, δ-HCH, p,p′-DDE, p,p′-DDD, and p,p′-DDT. An earlier study in Kenya reported that α-HCH and γ-HCH pesticides are still being used for crop protection [20]. This agricultural practice may have contributed to their occurrence in the soil samples. On the other hand, DDT was prohibited for use, but, due to its persistence, nature residues from historical input could still be present [45]. Overall, the potential sources of OCPs in the studied areas were attributed to the use of pesticides for agriculture and malaria control.

### 3.3. The Relationship between DDTs and HCHs Levels and TOC

OCPs have hydrophobic properties, and this renders them readily adsorbed by soil organic matter. Therefore, to show the relation between DDTs, HCHs, and total organic carbon (TOC), the Pearson correlation test was done (Table 4). The results depicted an association between TOC and OCPs, and amongst the DDT and HCH compounds in all the study sites. In Nyeri and Kapsabet, TOC revealed a positive relationship with α-HCH (P = 0.506) and β-HCH (P = 0.470), respectively, at the 0.01 significant level. Studies show that β-HCH adsorbs readily into organic matter and is, therefore, very persistent and does not readily evaporate from the soil [46]. Additionally, in Voi, TOC showed a significant positive relationship with p,p′-DDD (P = 0.971) at the 0.05 significance level. This association could imply that TOC enhances the assimilation of both HCH and DDT compounds in the soil. Hitherto, an association has been reported between organic contaminants and organic carbon where organic matter increases the transfer of OCPs in soil [25] and also provides carbon that promotes microbial degradation of OCPs [6]. Moreover, even though the OCP concentration varied throughout the three study sites, a significant association was depicted between them. The association could indicate that HCHs and DDTs may have emanated from similar sources. Analogies drawn from the principal component analysis revealed the same assumptions that suggest the utilization of agro-based pesticides and the use of insecticides to control malaria as the common source of OCPs in the soil.

### 3.4. Health Risk Assessment 

The incremental lifetime cancer risk (ILCR) technique was employed to estimate the human health risk from interaction with DDTs and HCH. This technique highlights that there a potential danger for human beings when exposed to OCPs via three major routes, including ingestion, inhalation, and dermal contact [47]. This model also indicates that an ILCR between 1 × 10^−6^ and 1 × 10^−4^ indicated probable danger, an ILCR of 1 × 10^−6^ or less signified immunity, whereas an increased possibility of a threat to human health was expected by ILCR value of more than 1 × 10^−4^ [47,48]. In this study, ILCRs average values were 8.70 × 10^−7^, 5.45 × 10^−7^, and 8.94 × 10^−7^ for children, adolescents, and adults, respectively (Table 5). The obtained ILCR cumulative values did not go above the set mark of 1 × 10^−6^ but were close signifying that the probability of cancer risk from soil particles containing HCHs and DDTs was acceptable, but not low enough.

## 4. Conclusions

In the present study, the restricted organochlorine pesticide (DDTs and HCHs) compounds were still detected in the soil, possibly due to their continued use and persistence in nature. The dominant OCPs in this study were HCHs, while DDTs were present at significantly lower concentrations. Kapsabet town exhibited the highest OCP contamination with a maximum of 52.7 ng/g. The DDTs probable source was particularly linked to the control of malaria and crop pests, whereas HCHs source was associated with the application of lindane for seed treatment. According to the risk assessment indices, the ILCR cumulative values obtained (10^−7^) were close to the target value (10^−6^) implying that the probability of human health risk from the sampled soils was not low enough and there was a need for control measures to be established. In comparison to previous studies, the soil pollution levels by OCPs for the three sampled areas can be quantified as slightly contaminated. However, a set of guidelines should be put in place to mitigate and control the use of OCPs in soils. The environmental authority in Kenya should also ensure that the prohibited OCPs are no longer in use.

## Figures and Tables

**Figure 1 ijerph-16-02937-f001:**
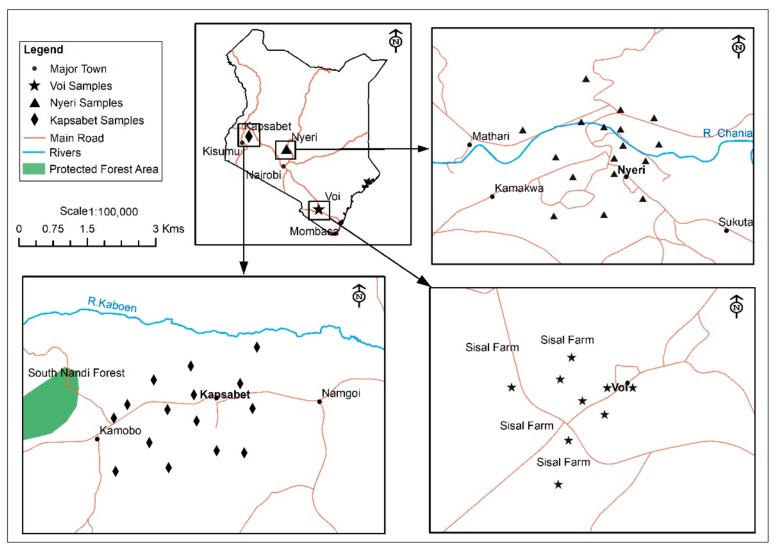
Map illustrating the location of sampling areas; Nyeri town in central highlands province, Kapsabet in Rift valley province, and Voi town in coast province.

**Figure 2 ijerph-16-02937-f002:**
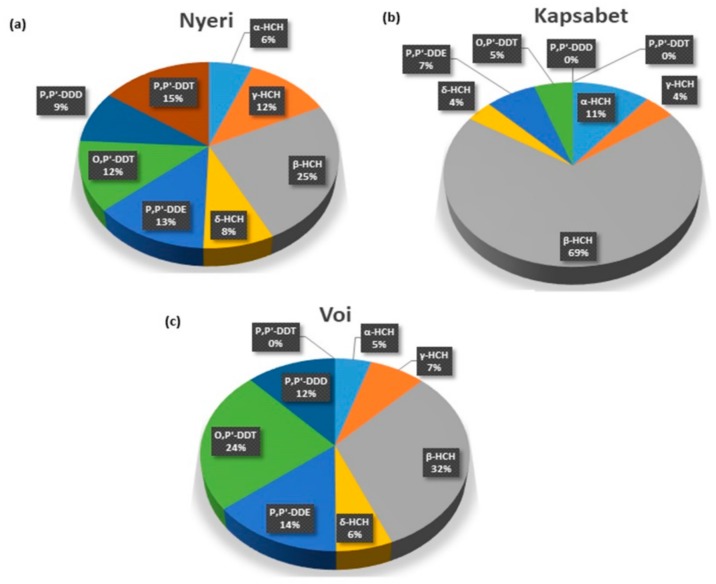
Pie charts showing the proportions and spatial distribution of DDTs and HCHs concentrations in (**a**) Nyeri, (**b**) Kapsabet, and (**c**) Voi towns in Kenya.

**Table 1 ijerph-16-02937-t001:** The parameters used in calculating the incremental cancer risk (ILCR) for humans exposed to environmental pollutants.

OCPs	$\mathrm{CS}F_{\mathrm{Ingestion}}$	$\mathrm{CS}F_{\mathrm{dermal}}$	$\mathrm{CS}F_{\mathrm{Inhalation}}$
α-HCH	6.30	4.49	6.30
β-HCH	1.80	1.98	1.86
γ-HCH	1.30	1.34	1.80
δ-HCH	1.80	N/A	1.80
p,p′-DDE	3.40 × 10^−1^	4.86 × 10^−1^	N/A
p,p′-DDD	2.40 × 10^−1^	3.43 × 10^−1^	N/A
p,p′-DDT	3.40 × 10^−1^	4.86 × 10^−1^	3.40 × 10^−1^
o,p′-DDT	N/A	N/A	N/A

N/A: not available. Source: [25].

**Table 2 ijerph-16-02937-t002:** The concentrations of HCHs, DDTs (ng/g), and TOC (%) in suburban soils from Nyeri, Kapsabet, and Voi. OCPs were reported in all the sampled locations.

OCPs	Nyeri (n = 23)			Kapsabet (n = 20)			Voi (n = 9)			
	Range	Mean ± Std. Deviation	Sum of Statistics	Range	Mean ± Std. Deviation	Sum of Statistics	Range	Mean ± Std. Deviation	Sum of Statistics	Dutch Standard Limits [32]
α-HCH	n.d.–0.62	0.20 ± 0.23	4.6	0.03–7.52	0.75 ± 1.7	14.9	n.d.–0.75	0.30 ± 0.3	2.73	17,000
γ-HCH	n.d.–1.32	0.39 ± 0.36	9.07	n.d.–0.48	0.30 ± 0.16	5.95	0.06–1.03	0.46 ± 0.3	4.16	1200
β-HCH	0.24–2.12	0.83 ± 0.56	19.2	n.d.–37.8	4.82 ± 10	96.3	0.02–4.16	1.99 ± 1.5	18.0	1600
δ-HCH	n.d.–0.66	0.27 ± 0.23	6.19	n.d.–2.25	0.25 ± 0.5	5.09	n.d.–0.92	0.39 ± 0.3	3.51	-
∑HCHs	0.24–4.72	1.69 ± 1.4	39.1	0.03–48.1	6.12 ± 13	122	0.06–6.86	3.14 ± 2.4	28.4	-
α-HCH/γ-HCH	-	0.51	-	-	2.51	-	-	0.66	-	
p,p′-DDE	n.d.–2.16	0.44 ± 0.50	10.1	n.d.–1.66	0.48 ± 0.44	9.65	n.d.–3.3	0.91 ± 0.97	8.16	2300
o,p′-DDT	n.d.–2.87	0.41 ± 0.95	9.49	n.d.–3.02	0.37 ± 0.90	7.41	n.d.–8.75	1.52 ± 3.2	13.6	1700
p,p′-DDD	n.d.–2.92	0.30 ± 0.85	6.93	n.d.	n.d.	n.d.	n.d.–3.46	0.75 ± 1.5	6.71	34,000
p,p′-DDT	n.d.–11.6	0.51 ± 2.4	11.6	n.d.	n.d.	n.d.	n.d.	n.d.	n.d.	1700
∑DDTs	n.d.–19.6	1.66 ± 4.7	38.1	n.d.–4.68	0.85 ± 1.3	17.1	n.d.–15.5	3.18 ± 5.6	28.5	-
p,p′-DDE/p,p′-DDD	-	0.82	-	-	n.d.	-	-	0.84	-	
TOC	0.21–2.65	0.90–0.8	20.7	0.03–1.71	0.54–0.51	10.7	0.25–1.25	0.56–0.38	5.0	
∑OCPs	0.24–24.3	3.35 ± 6.1		0.03–52.7	6.97 ± 14		0.06–22.4	6.32 ± 8.02		

n.d.: Concentrations below the limit of detection; OCPs: Organochloride Pesticides; HCHs: hexachlorocyclohexane; DDTs: Dichlorodiphenyltrichloroethane; DDD: dichlorodiphenyldichloroethane; DDE: dichlorodiphenyldichloroethylene; TOC: total organic carbon.

**Table 3 ijerph-16-02937-t003:** Factor loadings for principal rotated component matrix for DDTs and HCHs in Nyeri, Kapsabet, and Voi towns.

OCPs	Nyeri				Kapsabet			Voi	
	PC1	PC2	PC3	PC4	PC1	PC2	PC3	PC1	PC2
α-HCH	0.324	0.493	−0.687	0.176	0.927	−0.106	−0.041	0.699	0.561
β-HCH	0.830	−0.114	0.081	−0.233	−0.681	−0.355	0.448	0.803	0.587
γ-HCH	0.648	−0.266	−0.229	0.439	0.967	−0.065	0.021	0.854	0.306
δ-HCH	0.208	0.093	0.913	0.166	−0.215	−0.198	−0.717	0.649	0.555
p,p′-DDE	−0.204	0.039	0.106	0.918	−0.373	−0.287	0.701	0.579	0.755
p,p′-DDD	−0.168	0.889	−0.114	−0.118	−0.131	−0.067	0.732	0.400	0.816
p,p′-DDT	−0.051	0.910	0.056	0.099	−0.026	0.993	−0.029	−0.054	0.82
o,p′-DDT	0.836	0.019	0.067	−0.101	−0.026	0.993	−0.029	−0.765	0.074
Eigenvalues	2.29	1.83	1.27	1.17	2.89	2.13	1.42	5.33	1.02
% of variance	28.6	22.9	15.9	14.7	36.2	26.6	17.6	66.5	12.7
Cumulative %	28.6	51.5	67.4	82.0	36.2	62.9	80.6	66.6	79.3

**Table 4 ijerph-16-02937-t004:** Pearson correlation matrix illustrating the TOC, DDTs, and HCHs concentrations, in suburban soils from Nyeri, Kapsabet, and Voi towns.

OCPs	TOC	α-HCH	γ-HCH	β-HCH	δ-HCH	p,p′-DDE	o,p′-DDT	p,p′-DDD	p,p′-DDT
Nyeri									
TOC	1	0.506 *	0.377	0.240	−0.153	−0.077	0.270	0.094	0.066
α-HCH		1	0.154	0.212	−0.402	0.05	0.386	0.351	0.167
γ-HCH			1	0.353	0.176	−0.305	−0.271	−0.132	0.559 **
β-HCH				1	0.003	0.096	−0.286	−0.203	0.384
δ-HCH					1	0.159	−0.066	0.088	0.151
p,p′-DDE						1	−0.063	0.116	−0.191
o,p′-DDT							1	0.696 **	−0.095
p,p′-DDD								1	−0.077
p,p′-DDT									1
Kapsabet									
TOC	1	0.289	−0.059	0.470 *	−0.161	−0.236	−0.076	−0.131	−0.131
α-HCH		1	−0.585 **	0.842 **	−0.065	−0.29	−0.12	−0.101	−0.101
γ-HCH			1	−0.622 **	−0.285	0.534 *	0.268	−0.353	−0.353
β-HCH				1	−0.208	−0.322	−0.12	−0.09	−0.09
δ-HCH					1	−0.18	−0.147	−0.12	−0.12
p,p′-DDE						1	0.576 **	−0.257	−0.257
o,p′-DDT							1	−0.067	−0.067
p,p′-DDD								1	1.00 **
p,p′-DDT									1
Voi									
TOC	1	0.447	0.406	0.126	0.289	0.514	0.471	0.971 **	−0.24
α-HCH		1	0.871 **	0.874 **	0.646	0.756 *	0.664	0.473	−0.412
γ-HCH			1	0.866 **	0.872 **	0.929 **	0.794 *	0.412	−0.543
β-HCH				1	0.603	0.705 *	0.606	0.134	−0.441
δ-HCH					1	0.801 **	0.696 *	0.354	−0.442
p,p′-DDE						1	0.889 **	0.495	−0.35
o,p′-DDT							1	0.488	−0.18
p,p′-DDD								1	−0.189
p,p′-DDT									1

** Correlation is significant at the 0.01 level; * Correlation is significant at the 0.05 level.

**Table 5 ijerph-16-02937-t005:** Cancer risk potential values due to ingestion, dermal contact, and inhalation of DDTs and HCHs in suburban soils of Nyeri, Kapsabet, and Voi towns.

Growth stage	Exposure Pathways	Maximum	Mean	Median
Childhood	Ingestion	3.09 × 10^−6^	7.22 × 10^−7^	2.21 × 10^−7^
	Dermal contact	7.55 × 10^−7^	1.48 × 10^−7^	2.74 × 10^−8^
	Inhalation	1.28 ×10^−10^	2.92 × 10^−11^	7.88 × 10^−12^
	Cancer risk	3.85 × 10^−6^	8.70 × 10^−7^	2.49 × 10^−7^
Adolescence	Ingestion	1.61 × 10^−6^	3.76 × 10^−7^	1.15 × 10^−7^
	Dermal contact	8.66 × 10^−7^	1.69 × 10^−7^	3.14 × 10^−8^
	Inhalation	2.17 ×10^−10^	4.95 × 10^−11^	1.33 × 10^−11^
	Cancer risk	2.48 × 10^−6^	5.45 × 10^−7^	1.47 × 10^−7^
Adult	Ingestion	2.96 × 10^−6^	6.92 × 10^−7^	2.12 × 10^−7^
	Dermal contact	1.03 × 10^−6^	2.02 × 10^−7^	3.75 × 10^−8^
	Inhalation	3.94 × 10^−10^	9.00 × 10^−7^	2.43 × 10^−11^
	Cancer risk	4.00 × 10^−6^	8.94 × 10^−7^	2.50 × 10^−7^

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
