# Peer review of "Occurrence and Toxicological Risk Evaluation of Organochlorine Pesticides from Suburban Soils of Kenya"

_ijerph, 2019, doi:10.3390/ijerph16162937_

Round 1

Reviewer 1 Report

I think this manuscript is very interesting, and provides information on a field that is being forgotten in developed countries. I think it is interesting to know that OCs are still being used in some countries, and their potential health and environmental risks. 

I only have some minor points that I would like the authors to address:

1) The source of analytical standards is not mentioned in the text.

2) Detection limits are not given.

3) There is some confusion about wether the ECD or the MS detectors were used. In the abstract a "gas chromatography electron capture detector" is mentioned (lines 18 and 19), but later on a GC-ECD-MS is mentioned.

Author Response

Reviewer 1 Comments and Suggestions for Authors

I think this manuscript is very interesting, and provides information on a field that is being forgotten in developed countries. I think it is interesting to know that OCs are still being used in some countries, and their potential health and environmental risks.

I only have some minor points that I would like the authors to address:

1) The source of analytical standards is not mentioned in the text.

Response: Thank you for the comment. The source of analytical standards used were mentioned as advised.

2) Detection limits are not given.

Response: The detection limits were included in the manuscript text.

3) There is some confusion about wether the ECD or the MS detectors were used. In the abstract a "gas chromatography electron capture detector" is mentioned (lines 18 and 19), but later on a GC-ECD-MS is mentioned.

Response: Thank you for the comment. A GC-ECD was used and a correct name of the GC-machine was given in the manuscript text. 

Reviewer 2 Report

The subject is interesting and methodology is in accordance with modern trends in analytical chemistry, but in my opinion there are not too much novelty, it is typical monitoring (but interesting) work.

There is a lack of applied method validation. Where are there the LOD, LOQ, precision and accuracy? Were there used a reference materials, interlaboratory comparisons?

There are too much significant figures in data. Results should be expressed with 3 significant figures (at most) - it appears from uncertainty, and SD with two significant figures.

v. 113 - incorrect low indexes

v. 307-312 - incorrect high indexes

Author Response

Reviewer 2

The subject is interesting and methodology is in accordance with modern trends in analytical chemistry, but in my opinion there are not too much novelty, it is typical monitoring (but interesting) work.

There is a lack of applied method validation. Where are there the LOD, LOQ, precision and accuracy? Were there used a reference materials, interlaboratory comparisons?

 Response: The method validation was included in the manuscript text under Materials and method, section 2.5 Quality control.

There are too much significant figures in data. Results should be expressed with 3 significant figures (at most) - it appears from uncertainty, and SD with two significant figures.

Response: Thank you for the comment. The results data was revised and expressed using relevant significant figures as advised.

v. 113 - incorrect low indexes

Response: The low indexes were corrected.

v. 307-312 - incorrect high indexes

Response: The high indexes were corrected.

Reviewer 3 Report

The manuscript reveals the distribution of organochlorine pesticides, residual concentrations in soil, and potential risks to the environment and human health by detecting organochlorine pesticides in the soils of suburbs of Kenya. This study has merit, however, it requires some improvements.

1.    It is advisable that some information and references contained in the introduction and discussion be updated.

2.    It is recommended to quote toxicological data for organochlorine pesticides and compare them with the test data.

3.    The relevant articles published recently by IJERPH are not cited.

Author Response

Reviewer 3

The manuscript reveals the distribution of organochlorine pesticides, residual concentrations in soil, and potential risks to the environment and human health by detecting organochlorine pesticides in the soils of suburbs of Kenya. This study has merit, however, it requires some improvements.

1.    It is advisable that some information and references contained in the introduction and discussion be updated.

Response: Thank you for the suggestion. The information and references were updated as advised.

2.    It is recommended to quote toxicological data for organochlorine pesticides and compare them with the test data.

Response: Thank you for the suggestion. Data on OCPs in our study was compared with set toxicological standard values for Dutch soils. The information was included in the manuscript text in the results and discussion, section 3.1.

3.    The relevant articles published recently by IJERPH are not cited.

Response: Thank you for the comment. The IJERPH relevant published articles were included as suggested.

References that were added during revision

Gereslassie, T.; Workineh, A.; Atieno, O. J.; Wang, J. Determination of Occurrences, Distribution, Health Impacts of Organochlorine Pesticides in Soils of Central China. Int J Environ Res Public Health 2019, 16, 146.

Kim, Y. A.; Park, J. B.; Woo, M. S.; Lee, S. Y.; Kim, H. Y.; Yoo, Y. H. Persistent Organic Pollutant-Mediated Insulin Resistance. Int J Environ Res Public Health 2019, 16, 448.

Ray, D., Nervous system and behavioral  toxicology. In Comprehensive Toxicology, 2 ed.; McQueen, C. A., Eds. Harrison School of Pharmacy, Auburn University: Auburn, AL, USA, 2010; Vol. 2. pp.8624.

Qu, C.; Qi, S.; Dan, Y.; Huang, H.; Xing, X. Risk assessment and influence factors of organochlorine pesticides (OCPs) in agricultural soils of the hill region A case study from Ningde, southeast ChinaOriginal. J Geochem Explor 2015, 149, 43-51.

Cui, L.; Wei, L.; Wang, J. Residues of organochlorine pesticides in surface water of a megacity in central China: seasonal-spatial distribution and fate in Wuhan. Environ Sci Pollut Res 2016, 24, 1-12.

Phillips, T. M.; Seech, A. G.; Lee, H.; Trevors, J. T. Biodegradation of hexachlorocyclohexane (HCH) by microorganisms. Biodegradation 2005, 16, 363-392.

Netherlands Ministry of Housing, Spatial planning and environment's circular on target values and intervention values for soil remediation. Available online: http://www2.minvrom.nl/Docs/internationaal/annexS_I2000.pdf ( Accessed on 27 July 2019).

Yahaya, A.; Okoh, O. O.; Okoh, A. I.; Adeniji, A. O. Occurrences of Organochlorine Pesticides along the Course of the Buffalo River in the Eastern Cape of South Africa and Its Health Implications. Int J Environ Res Public Health 2017, 14, 1372.

Round 2

Reviewer 2 Report

After correction it can be accepted

Reviewer 3 Report

The authors have answered all questions of my review, so the revised manuscripts is satisfied. Therefore, I recommended to be published.